# Breast Radiotherapy after Oncoplastic Surgery—A Multidisciplinary Approach

**DOI:** 10.3390/cancers14071685

**Published:** 2022-03-25

**Authors:** Gabrielle Metz, Kylie Snook, Samriti Sood, Sally Baron-Hay, Andrew Spillane, Gillian Lamoury, Susan Carroll

**Affiliations:** 1Northern Sydney Cancer Centre, Royal North Shore Hospital, Sydney, NSW 2065, Australia; samriti.sood@health.nsw.gov.au (S.S.); sbaronha@med.usyd.edu.au (S.B.-H.); gillian.lamoury@health.nsw.gov.au (G.L.); susan.carroll@health.nsw.gov.au (S.C.); 2Sydney Medical School, University of Sydney, Sydney, NSW 2006, Australia; kylie.snook@surgicaloncology.com.au (K.S.); andrew.spillane@melanoma.org.au (A.S.); 3Breast and Surgical Oncology, The Poche Centre, Sydney, NSW 2060, Australia; 4The Mater Hospital, Sydney, NSW 2060, Australia; 5Breast and Melanoma Surgery Unit, Royal North Shore Hospital, Sydney, NSW 2065, Australia

**Keywords:** breast cancer, radiation therapy, oncoplastic breast surgery

## Abstract

**Simple Summary:**

This article aims to review and summarize the current evidence for the role of oncoplastic breast surgery and the implications this may have on other therapies, such as radiotherapy and chemotherapy.

**Abstract:**

Oncoplastic breast surgery encompasses a range of techniques used to provide equitable oncological outcomes compared with standard breast surgery while, simultaneously, prioritizing aesthetic outcomes. While the outcomes of oncoplastic breast surgery are promising, it can add an extra complexity to the treatment paradigm of breast cancer and impact on decision-making surrounding adjuvant therapies, like chemotherapy and radiotherapy. As such, early discussions at the multidisciplinary team meeting with surgeons, medical oncologists, and radiation oncologists present, should be encouraged to facilitate best patient care.

## 1. Introduction

Modern breast cancer treatment is rapidly evolving with shifts in treatment paradigms aiming to continuously improve outcomes for patients. The overarching aim of treatment is to obtain optimal oncological results for patients. As a consequence of improvements in systemic therapy, more precise surgical procedures and radiation techniques, as well as early detection with screening programs, patients are living longer with better quality of life (QOL). The increasing importance of incorporating psychosocial elements and QOL into treatment decision-making parallels the advances in breast cancer survival.

Oncoplastic and reconstructive breast surgery describes breast surgery that prioritizes both oncologic and cosmetic outcomes. Advancement in surgical techniques over the past decades has led to more widespread emergence of the use of these techniques which can avoid aesthetic deformity, even in locally advanced disease [1].

The complexity of this approach requires multidisciplinary planning from the outset to facilitate appropriate integration of neoadjuvant and adjuvant therapy. Given it is well established that adjuvant radiotherapy is usually required after breast conservation and as the indications for post-mastectomy radiotherapy become more established, the need for appropriate inclusion of the radiation oncologist in early discussions is imperative. [2] Additionally, as the trend towards a neoadjuvant approach for systemic therapy is increasing and many patients with early-stage breast cancer are being offered chemotherapy, medical oncologists need to be incorporated into discussion, too.

We aim to highlight the importance of a multidisciplinary integration of oncoplastic and reconstructive surgical techniques to breast cancer management in the modern era.

## 2. Defining Oncoplastic Surgery

There have been multiple attempts by various groups to define oncoplastic breast surgery over recent years [3,4]. While there is no universal consensus definition, the overall aim of oncoplastic surgery remains similar, which involves utilizing surgical techniques to restore symmetry, preserve normal tissue, optimize cosmesis, and minimize post-operative complications while simultaneously achieving excellent oncological outcomes [4,5,6]. The incorporation of oncoplasty with breast-conserving surgery can serve many purposes; to fill the defect associated with the volume loss of a lumpectomy, to perform a reduction lift procedure for reasons of cosmesis and/or comfort, and to produce greater symmetry in the contralateral breast [7]. Oncoplastic breast surgery incorporates more optimal incision placement (often via mastopexy, circumareolar, or inframammary fold incisions) for superior cosmetic outcomes. Historically, breast reduction has also been performed for better tolerance of radiotherapy [8,9] however with modern breast radiotherapy techniques, recent publications suggest that breast size does not have any implication on late radiotherapy effects [10,11].

The American Society of Breast Surgeons recently published a consensus statement regarding the definition and classification of oncoplastic surgery [12]. This has been defined as “breast-conservation surgery incorporating an oncoplastic partial mastectomy with ipsilateral defect repair using volume displacement or volume replacement techniques with contralateral symmetry surgery as appropriate”.

Another way to define oncoplastic surgery can be via categorising oncoplastic surgical techniques according to differing skill levels. Clough et al. attempt to simplify this by proposing two levels in the classification of oncoplastic breast surgery. Level 1 refers to procedures needing less than 20% reduction of breast tissue and level 2 refers to procedures needing more than 20% volume resection. Level 2 procedures are more complex and split into volume displacement and volume replacement techniques. Volume displacement procedures use mammoplasty techniques to fill the glandular defect. Volume replacement techniques refer to procedures where the glandular gap is filled with either autologous tissue from another site or with an implant [4].

Oncoplastic surgery is also considered relevant in women undergoing mastectomy who elect for breast reconstruction. After mastectomy, there are multiple reconstructive options which can be classified by the timing of reconstruction, i.e., immediate or delayed or the type of surgical procedure including prosthetic reconstruction (tissue or expander implant), local flap reconstruction, and distant pedicled or free flap reconstruction, which includes transverse rectus abdominis (TRAM) flap, deep inferior epigastric perforator (DIEP) flap, or latissimus dorsi (LD) flap [7].

Whilst traditionally a team approach with breast and plastic surgeons would have always been required for these procedures, many breast surgery training programmes now incorporate oncoplastic breast surgery into their curricula, such that many contemporary breast surgeons now perform level 1 and level 2 oncoplastic procedures as well as some reconstructive procedures [13]. Alternatively, a team approach with breast and plastic surgeon is still used by some and is still usually required for autologous free flap procedures.

## 3. Role of Surgery and the Surgeon 

### 3.1. Identifying Eligible Patients 

All patients should be considered for oncoplastic surgery [14,15]. The type of surgery most appropriate for individual patients will be dependent on the patient’s age, tumour features, breast size and shape, and surgical skill. Other important patient considerations are local and systemic disease burden, prior or planned local radiotherapy, genetic risk factors, co-morbidities, smoking history, and impact of recovery time on employment and lifestyle [16]. 

Interestingly, implementation of immediate reconstruction varies significantly on the basis of socioeconomic status. SEER data has shown that older women with non-English-speaking background were less likely to have discussions about reconstructive surgery and less likely to undergo reconstruction. This trend has also been shown in recently published Australian data [17]. In addition, race, income level, education, and insurance type have been found to strongly associate with immediate reconstruction [7,18]. 

All suitable patients should be offered equal access to discussion about oncoplastic and reconstructive breast surgery given the reported benefits discussed below. 

### 3.2. Types of Oncoplastic Surgery 

#### 3.2.1. Breast-Conservation Setting

The goal of breast-conservation surgery is to remove the tumour with adequate margins and achieve acceptable cosmetic outcomes. This has become the standard of care since the landmark EBCTC paper showing equivalence of breast conservation and mastectomy [19,20] and recent Swedish cohort studies have shown improvements in breast cancer specific survival in women who have breast-conservation surgery with radiotherapy versus mastectomy [21]. The use of breast-conservation surgery continues to increase as cancers are diagnosed at earlier stages and are more suitable to avoid mastectomy. This can be attributed to implementation of screening programs but also shrinkage of tumour that is obtained with neoadjuvant systemic therapy, especially in more recent times with increasing use of neoadjuvant chemotherapy. 

Aesthetic success in breast conversation surgery is dependent upon a variety of patient and tumour-specific factors and, unfortunately, 25–30% of patients experience significant deformity following surgery [22]. Small breast size, ptotic breast shape, large body habitus, large tumour size, central, medial, or lower quadrant tumour location, segmental or multifocal tumour distribution, tumour re-excision, and resection of >20% breast volume have all been identified as predictors of poor cosmesis [23,24,25]. Post-operative breast asymmetry after breast-conserving surgery also shown to correlate with poor psychosocial outcomes [26].

Conventional lumpectomy generally removes 20–40 g of breast tissue and it has been postulated that adverse cosmetic outcomes are seen when more than 80 g of breast tissue is removed. Depending on the patient, oncoplastic surgical techniques allow for up to 200–1000 g of breast tissue to be removed without causing significant cosmetic deformity due to the reconstructive techniques applied.

For these reasons, the use of oncoplastic surgery in the setting of breast conservation is increasing. For example, MD Anderson Cancer Centre data reported an increase from 4% to 15% in those who had breast-conserving surgery coupled with oncoplastic techniques for breast restoration [15,27].

The various types of oncoplastic surgery available in the breast-conservation setting are summarized in Table 1 [4].

#### 3.2.2. In the Mastectomy Setting

Simple mastectomy has traditionally been the standard of care for patients undergoing mastectomy until the introduction of skin sparing mastectomy which was first described in 1991 by Toth and Labbert. This involves removal all breast glandular tissue while leaving a skin envelope +/− the nipple–areolar complex (NAC) to achieve a more natural aesthetic outcome [28]. Consensus states there is no oncological difference between these approaches in regards to local recurrence rates and survival, however, large randomized trials in this area are lacking [29,30]. 

Immediate or delayed breast reconstruction should be considered in all patients who are recommended or choose to have mastectomy. Optimal timing and sequencing of reconstruction is controversial in the context of neoadjuvant and adjuvant therapies which is explored later in this review. There has been a well-established trend towards increased usage of immediate reconstruction over the last decade. The National Cancer Database found that immediate reconstruction rates increased to 48% of all mastectomies in 2013 among women age younger than 75 years with early-stage breast cancer [31]. More recent registry data has shown a further increase in immediate implant reconstruction from 30% to 54% and immediate flap reconstruction from 17% to 21% [32]. 

Breast reconstruction can be performed using implantable breast prostheses, the patient’s own tissue (autologous flap), autologous fat grafting or a combination of techniques:,

##### Implant-Based Reconstruction

Implant-based techniques involve using a round or bi-dimensional breast-shaped silicone or saline-filled prosthesis to mimic the breast mound. Breast reconstruction can be performed as a single-stage procedure with immediate implant-based reconstruction or as a two-stage procedure with a temporary expandable prosthesis that is replaced at a later date with a permanent prosthesis. 

Immediate implant-based reconstruction is often referred to as a ‘direct-to-implant’ surgery and involves insertion of a permanent implant into the skin enveloped in a single procedure. This is often pre-pectoral and assisted by use of acellular dermal matrices or other mesh products. In many centres this has become the dominant form of implant-based reconstruction in recent times.

A two-stage implant-based reconstruction refers to insertion of a temporary implant or tissue expander which is either placed completely sub-pectoral or in the prepectoral space using acellular dermal matrix or other mesh product. A tissue expander is sometimes preferred if post-mastectomy radiotherapy is indicated as the expander can be inflated/deflated to allow for radiotherapy planning. It also commits the patient to a second procedure which some feel necessary when radiotherapy is given. At the time of surgery, if the skin flaps appear to have suboptimal blood supply, the placement of an underfilled tissue expander may be preferred to prevent skin necrosis, by allowing skin flap perfusion to be optimised, rather than under pressure from a large permanent implant. Two stage implant reconstruction may also be some surgeons’ preference in the setting of any implant-based reconstructions. 

##### Autologous Breast Reconstruction

Autologous breast reconstruction can similarly be done as an immediate or a delayed procedure. The delayed surgery can be done years after the initial surgery and tends not to be impacted by the delivery of radiotherapy [33].

Autologous reconstruction is performed with the use of a pedicled or free flap utilizing various donor sites to harvest skin and fat tissue. Most commonly, flaps are created from the abdominal wall and include the deep inferior epigastric artery perforator (DIEP) flat and the transverse rectum abdominis myocutaneous (TRAM) flap. Flaps from the back can also be utilized includes the latissimus dorsi (LD) flap. Flaps can also be taken from the gluteal region including the gluteal musculocutaneous and/or perforator flaps (SGAP and IGAP) and upper thigh with the transverse gracilis (TUG) flaps [34,35].

There is a paucity of randomised data to support which is the most optimal procedure and the origin of autologous tissue is dependent on patient’s body habitus including BMI, excess skin, shape of breast, and shape of chest wall. It also depends on patient preference, relevant medical history, surgeon preference and expertise, the type and extent of mastectomy, and extent of tissue needed for reconstruction [35].

A summary of the three most common autologous reconstructive options, their suitability and possible complications is included in Table 2. 

##### Autologous Fat Grafting

Autologous fat grafting (AFG) where fat cells are harvested with liposuction, prepared, and then injected into the body is commonly utilized as an adjunct to enhance cosmetic and functional outcomes in breast reconstruction [36]. AFG as a stand-alone procedure for breast reconstruction is in its infancy but there is emerging data to support this technique [37]. AFG can sometimes be associated with unwanted effects such a fat necrosis and oil cyst development which can impact cosmesis in terms of texture, breast contour, and abnormalities on subsequent breast imaging [38,39].

There is no correct technique for breast reconstruction but rather preferences according to patient, cancer treatment required, operative findings, and surgeon’s preference. Integration into an MDT discussion should occur to ensure all factors are considered prior to decision-making with a shared care model between the surgeon, patient, and medical and radiation oncologist.

## 4. Oncoplastic Outcomes 

Literature published regarding the oncological outcomes in oncoplastic surgery appear to be excellent with results paralleling non-oncoplastic breast surgery. However, it is important to note that validation in robust studies that constitute high levels of evidence is lacking. Despite this, oncoplastic surgery has been widely accepted as a suitable option across the globe and reasons for this are outlined below [40]. 

### 4.1. Margin Positivity

Involved surgical margins appear to occur less often when oncoplastic techniques are utilised. Recently published guidelines by an expert consensus panel included a meta-analysis of the current literature in oncological outcomes in oncoplastic breast conservation. On assessment of margin positivity, a well-established risk factor for local recurrence, a total of 13 studies reported data on margin positivity rates. They included more than 2000 patients with oncoplastic breast surgery and 5756 patients with standard breast-conservation surgery. Positive margins were reported in 6.2% of patients in the oncoplastic group versus 10.5% of patients in the standard surgical group. It should be noted that most of these were observational studies [41]. The same group reviewed the rates of margin re-excision from 12 observational studies which included more than 108,000 patients and determined the odds ratio for margin re-excision was 0.61 (95% CI 0.42–0.87) in favour of oncoplastic surgery [41].

This has been supported by three meta-analyses in recent years showing that regarding positive margins, oncoplastic breast surgery was non-inferior to conventional breast conservation [40,42,43].

Surgical clips should be placed at the lumpectomy site (breast parenchyma and chest wall) to delineate the surgical margins prior to oncoplastic rearrangement. This allows for accurate identification of surgical margins should re-excisional surgery be necessary. This also will aid the radiation oncologist to delineate a boost volume if clinically indicated. Cavity shaving of lumpectomy margins before oncoplastic techniques can be considered to reduce the incidence of close or involved margins [42]. 

### 4.2. Local Recurrence Rates

As survival outcomes for breast cancer patients tend to be exceptionally favourable, local control becomes increasingly relevant. To date, there has been no suggestion that oncoplastic surgery results in inferior local control rates compared to standard breast surgery.

Clough et al. published a study suggesting that the 5-year local recurrence rates in patients with oncoplastic breast-conservation surgery was only 2.2% [43].

De Lorenzi et al. published a case-matched cohort comparison study of larger cancers (T2) with a 7.4-year average follow-up period and found no statistically significant differences in local or regional recurrences between patients undergoing oncoplastic surgery and those undergoing mastectomy [44].

A systematic review of oncological outcomes after reduction mastopexy included 17 articles with 1324 patients. This group reported a pooled locoregional recurrence rate of 3.1% with 2-year follow-up. While this locoregional recurrence rates falls in line with historical data, the short follow up for this cohort of patients who have a long survival needs to be acknowledged. Additionally, most patients in this review had small tumours with less than 20% of patients having T3 tumours [45].

### 4.3. Conversion to Mastectomy

While strong retrospective data is lacking regarding conversion rates to mastectomy from oncoplastic breast-conserving surgery, Crown et al. recently published an observational cohort study of 100 patients with invasive breast cancer with multiple ipsilateral lesions and/or disease spanning >5 cm treated who underwent oncoplastic surgery at a single institution. Overall, 13% of patients converted to a mastectomy, however, importantly, 7% of these patients elected to have a mastectomy rather than re-excision [46].

Similarly, Heeg et al. published a population-based study looking at 13,185 patients who underwent standard breast-conservation therapy and 5003 patients who underwent oncoplastic surgery. Re-excision rates were 15.6% after breast-conserving therapy and 14.1% after oncoplastic surgery. After adjusting for confounders, patients were less likely to have re-excision following oncoplastic surgery than breast-conserving therapy specifically after volume displacement and reduction. This is possibly due to the larger volume of tissue that is excised. The rate of conversion to mastectomy was 3.2% after oncoplastic surgery and 3.7% after breast-conserving surgery after volume displacement and reduction procedures [47]. 

### 4.4. Overall Survival

There is a paucity of valid long–term data from high level evidence regarding overall survival in patients having oncoplastic breast surgery procedures. Carter et al. published a retrospective review that assessed oncological outcomes of 9861 patients who underwent oncoplastic reconstruction or other breast procedures. This showed that there was no difference in survival between oncoplastic breast conservation and standard breast-conservation surgery. Most patients had T1-T2 tumours but those patients who had reconstructive surgery were more likely to harbor node-positive disease (18.4 vs. 11.4%; *p* < 0.0001) [15]. 

### 4.5. Complication Rate

Complications unique to oncoplastic resections include nipple necrosis, fat necrosis, and delayed healing. There are many ways that these can be minimised, including staying in true anatomic planes, maintaining meticulous hemostasis, and avoiding overly aggressive undermining in patients with a fatty breast. Similar to any surgery, complication rates are higher in smokers, diabetics, and morbidly obese patients and general complications like bleeding and infection appear to be the same amongst oncoplastic versus non-oncoplastic patients. 

Unsurprisingly, compared with standard breast-conservation surgery, oncoplastic surgery tends to have a higher short-term complication rate. This is likely due to more extensive dissection and more delicate surgery that is needed during surgery with an oncoplastic design [48]. Jonczyk et al. evaluated this in a National Surgical Quality Improvement Program database analysis and demonstrated that the short-term overall complication rates for standard partial mastectomy were lower (2.25 %) than those for oncoplastic surgery (3.2%) [49].

Losken et al. compared 111 patients who underwent newer oncoplastic techniques with a standard oncoplastic group of 222 patients. These newer techniques used an extended pedicle (e.g., an extended superomedial pedicle) or a secondary pedicle (e.g., an inferolateral segment of the Wise incision in addition to a primary pedicle). No statistically significant difference in overall complication rates was observed between the oncoplastic-only group (15.5%), the extended pedicle group (19.6%), and the secondary pedicle group (20%) [50].

In addition to this, there tends to be an increase in operating theatre time needed for oncoplastic procedures, particularly if needing subsequent operations to adjust cosmesis. The impact of this on the patient but also on departments and workflow need to be considered. 

While it is expected that short-term complications may be slightly higher, it is important to note that complication rates tend to equalize in the long term. A 2014 meta-analysis supported findings highlighted that at three to five years, patients who underwent oncoplastic resections developed fewer complications (16 vs. 26%) and local recurrences (4 vs. 7%) and had a higher satisfaction with the appearance of their breast (90 vs. 83%) [43,51,52]. 

### 4.6. Patient Reported Outcomes

To date, there has been a paucity of data on patient-reported outcomes post oncoplastic surgery, however, this trend appears to be changing with more research recently published within this realm.

A recent publication including 48 patients evaluating patient reported outcomes post oncoplastic breast reconstruction using a post-surgery BREAST-Q score reported high satisfaction with overall outcome with an average score of 80.8% [53]. 

Interestingly, another recent publication including 1130 patients compared outcomes in patients with breast reconstruction or oncoplastic surgery. Patients in the oncoplastic group showed statistically significant higher psychosocial and aesthetic outcomes. Of note, these patients in the oncoplastic group had an earlier stage cancer. Similar, a recent Danish publication reported significantly better outcomes in Health-related Quality of Life in patients treated with oncoplastic breast surgery [54]_._


### 4.7. Surveillance 

In Australia, common practice in surveying patients after treatment for breast cancer include annual breast imaging with an ultrasound and mammogram in the breast-conservation setting or ultrasound only in the mastectomy setting. Importantly, the use of oncoplastic surgery does not seem to hinder the ability to screen patients with mammoplasty [51].

## 5. The Multidisciplinary Team

Multidisciplinary care can be broadly defined as an integrated team approach to healthcare in which medical and allied health professionals consider all relevant treatment options and develop individual treatment plans for each patient collaboratively. The multidisciplinary team (MDT) or multidisciplinary tumour board involves a representation of a broad range of cancer specialists with significant experience and knowledge in the field of a specific cancer-sub site to make recommendations regarding optimal care for cancer patients [5,52].

The purpose of the multidisciplinary team is to gather a variety of specialists and health personnel in a cohesive manner to discuss sensitive patient clinical situations to deliberate and determine best management options in accordance with relevant guidelines. The multidisciplinary team, in the setting of oncology, typically includes surgeons, radiation oncologists, medical oncologists, radiologists, pathologists, allied health members, and research and nursing staff. The purpose of including a variety of personnel is to ensure a holistic approach to the patient situation is obtained. Additionally, each specialist has a unique view and understanding of their own specialty and a deep understanding of the nuances that can be applied to specific patient situations. It is not possible for specialists to have specialist-level knowledge in every domain of cancer care and as such, having each specialist attend ensures the best possible outcome is being discussed and considered for each patient. 

Additionally, MDTs provide a unique opportunity for education and learning from other specialty groups. Another perceived benefit, which indirectly can impact patient outcomes, is the opportunity that MDTs provide to develop inter-personal skills, collegiality, build relationships and establish referral patterns amongst specialists. 

It is well established that multidisciplinary teams improve outcomes for oncology patients. Kesson et al. compared the impact of the introduction of multidisciplinary teams (MDTs) in a Scottish Health Board in Glasgow on patient survival with survival in surrounding regions that did not introduce MDTs for patients with breast cancer. They showed an improved survival and reduced variation in care in the region that introduced MDTs, in 13,722 women treated for breast cancer [55].

Prevention, screening, prompt diagnosis, and rapid referrals are critically important facets of every oncological management plan at a local, regional, national, and international level to improve cancer control and cancer outcomes. A well-established (MDT) can help facilitate this to occur in a smooth manner and, subsequently, continue to improve outcomes for patients [56].

Pre-operative MDTs also provide the opportunity to advise which type of surgery is most appropriate for individual patients, i.e., conserving the breast or mastectomy, identify cases prone to complications like smoking, diabetes, body habitus, estimate the risk of needing post-mastectomy radiotherapy prior to surgery, and estimate the risk versus benefit of post-mastectomy radiotherapy. Consideration of oncological options, including neoadjuvant therapy and/or oncoplastic techniques that may negate the need for a mastectomy or avoid massive defects in breast tissue, need to be explored within this setting. 

Importantly, patients feel reassured when they are aware their case has been discussed at an MDT and relieved to know that their treatment decision has been made by multiple specialists who have had the opportunity to hear about their specific case. 

Given complexities associated with incorporating oncoplastic surgery into the management paradigm, which will be explored below, we suggest that all new patients diagnosed with breast cancer be presented at the MDT prior to undertaking oncoplastic surgery [57].

## 6. The Role of Radiotherapy

Adjuvant radiotherapy plays a pivotal role in the setting of breast-conservation therapy [58,59,60]. In the well-renowned meta-analysis of over 10,000 women included in 17 randomised controlled trials, radiotherapy after breast-conservation therapy halves the rate of local recurrence and reduces cancer mortality by one sixth [61]. When mastectomy is required or elected, there are multiple indications for post-mastectomy radiotherapy to improve locoregional control and overall survival which will be discussed later. 

There are some important considerations in patients who undergo oncoplastic surgery who then may need adjuvant radiotherapy. 

A consensus statement has been published to assist with issues arising for radiation oncologists in patients after oncoplastic breast surgery. The statement highlights the importance of a multi-disciplinary approach and suggests that for optimal multidisciplinary collaboration, radiation oncologists should participate in or observe various types of oncoplastic procedures as part of their ongoing education. Other authors have also strongly suggested that surgeons arrange a preoperative referral to radiation oncology whenever the use of complex oncoplastic surgical techniques is anticipated. Similarly, the surgeons should also have a working knowledge of the techniques involved in the planning and delivery of radiation therapy. While not always achievable, the consensus panel suggests that surgeons attempt to be present with the radiation oncologists during target contouring and radiation boost planning. Through such partnerships, the challenge of translating geometric information from one medical specialty to another for the optimal treatment of the patient can be overcome [62]. 

### 6.1. Defining the Lumpectomy Cavity/Tumour Bed

In the setting of oncoplastic surgery, the ability to accurately delineate the tumour bed depends partially on the degree of tumour bed rearrangement that occurred at time of surgery. Defining the lumpectomy site is important for utilization of a boost dose in adjuvant radiotherapy but also for consideration of partial breast radiotherapy to ensure the at-risk tissue is adequately covered. 

The tumour bed can be difficult to localize post-oncoplastic surgery for multiple reasons; the post-operative scar is not useful in oncoplastic incisions as they often have no relationship to the tumour bed. Additionally, visualisation of a seroma has traditionally been used for tumour bed delineation, but movement of primary and secondary breast tissue pedicles in the setting of oncoplastic surgery can result in multiple seromas at varying sites [63]. When the nipple–areolar complex is rearranged, the tissue in each quadrant changes and the quadrants are defined relative to the nipple location. Similarly, glandular breast tissue flaps are often rotated from other areas of the breast into the lumpectomy cavity to fill the defect and true tumour bed margins can be well away from the original lumpectomy site [4,64].

Alco et al. published a prospective study evaluating the geographic variability of the tumour bed following oncoplastic surgery. A pre- and post-operative computerized tomography (CT) scan was obtained in 22 patients. Pre-operative volume was defined using the gross tumour volume or the biopsy tract and post-operative volume was delineated using surgical clips. The lumpectomy cavity shifted from the primary location in 36.4% with a median shift of 1.02 cm. Only 5 patients had pre and post-operative cavities that were fully aligned [65].

Pre-operative imaging can be utilized to assist with locating the tumour bed. Traditional breast-imaging modalities such as ultrasound and mammogram can be useful to describe the location of the tumour however magnetic resonance imaging (MRI) has been shown to have superior sensitivity and accuracy for detection and visualisation of tumour extent [66,67,68]. Fusion of the pre-operative MRI and the planning CT dataset can be used to further assist with the delineation of the tumour bed [69]. Utilising pre-operative MRI to assist with tumour bed delineation therefor can be considered helpful it is important to acknowledge that MRI is not always widely available for patients due to lack or resources or prohibitive costs. 

In addition to pre-operative imaging, insertion of surgical clips at the time of surgery is thought to assist with tumour bed delineation however the accuracy of this in the oncoplastic setting is yet to be determined. Aldosary et al. recently published data which used realistic breast phantoms to perform different oncoplastic breast surgical techniques and used two senior radiation oncologists to delineate the tumour bed on a CT data-set using the clips as standard protocol. The oncologist’s contours were compared to a ‘true tumour bed’ which was created via obtained CT images are regular phases of surgery to record pre- and post-closure surgical clip displacement. The post-surgical clips were significantly displaced from the original breast quadrant, indicating that surgical clips alone are not reliable surrogates of tumour bed location, highlighting the difficulty of tumour bed delineated post-oncoplastic surgery [70].

A specific scenario, not unique to oncoplastic surgery, is delineation of the tumour bed after neoadjuvant therapy. This is particularly relevant as there is an increasing trend towards a neoadjuvant therapy approach in breast cancer as highlighted elsewhere in this paper. In tumours expected to have dramatic responses to systemic therapy, like HER-2 positive cancers, the tumour bed can prove difficult to find for both the surgeon intra-operatively but also for the radiation oncologist attempting to define the tumour bed. For this reason, international breast cancer specialist panels have described the important of insertion of a radiopaque clip prior to commencing neoadjuvant systemic therapy [71]. In this scenario, a radiopaque marker is inserted, usually via ultrasound guidance, around the tumour prior to delivery of therapy which can then be located by the surgeon to allow adequate excision of tumour and/or reconstruction of the breast [72]. Some studies have shown that in 47% of cases, the radiopaque markers are the only remaining evidence of the original site of tumour [73]. While these radiopaque markers inserted pre-chemotherapy is helpful for the surgeon, clips should still be placed by the surgeon at the time of surgery to assist the radiation oncologist in delineating the surgical margins of the resected tumour bed. In addition to this, although not unique to oncoplastic surgery, expert consensus guidelines from GEC-ESTRO describe target delineation of the tumor bed in the context of partial breast radiotherapy where the tumor bed volume is of significant importance and these may be used to aid the radiation oncologist in contouring the tumor bed [74].

### 6.2. Dose Addition to the Tumour Bed

Delivering a radiation boost sequentially with adjuvant radiotherapy in the conserved breast is well documented in the literature as there is an established local control benefit in select patients however an overall survival benefit has not yet been shown [75,76,77]. There is emerging data for the higher dose-escalated boost to be delivered simultaneously via an intensity-modulated technique [78].

In order to accurately deliver a boost, the radiation oncologist must be able to readily identify the tumour cavity.

Garreffa et al. published a recent retrospective review where the tumour bed and boost volume were defined by the surgeon using surgical clips and the typical radiological appearance of the flap. The radiation oncologist delineated the boost as per standard procedures. Results were compared with a surgical specimen volume. There was no difference between the surgeons boost volume and the surgical specimen volume, however, the radiation oncologist’s volume was significantly smaller than the true surgical specimen. The groups approach by combining clips with the redefinition of the flap on CT may be a more accurate method to delineate the tumour bed [79].

Traditionally, the main concern with boosting the tumour bed has been a slight increase in fibrosis [80,81,82,83]. This is of particular concern in oncoplastic surgery where more tissue has been surgically manipulated which theoretically may increase the risk of fibrosis when a boost is utilized. It is important to remember that most patients with early stage, luminal A breast cancer have an excellent prognosis and the omission of a boost for a small local control benefit may be warranted in place of oncoplastic surgery. Conversely, young women may be most likely to opt for an oncoplastic procedure and younger women with large tumours have a higher risk of local recurrence. A boost may be of greatest benefit to these women however boosting large tumour beds is associated with higher rates of fibrosis and this may jeopardize the aesthetic outcome, the overall goal of oncoplastic breast surgery [52,67].

Given the complexities and controversies describes above, consensus guidelines for tumour bed localization for radiotherapy after oncoplastic surgery have recently been published. The main recommendations of the guidelines are:Surgical clips are necessary and should, at a minimum, be placed along the 4 side walls of the cavity, plus 1-4 clips are the posterior margin if necessary;Operative reports should include pertinent information to help guide the radiation oncologist;Breast surgeons and radiation oncologists should have a basic understanding of the oncoplastic surgical techniques and work on speaking a common language; andCareful consideration is needed when determining the value of targeted radiation, such as boost, in higher level oncoplastic surgical procedures with extensive tissue rearrangement [62].

For these reasons, the benefit of oncoplastic surgery needs to be considered in conjunction with the clinical indications for the type of adjuvant therapy and the implications this may have oncologically but also aesthetically. Given the complexities involved, we recommend this should be done with involvement of the multidisciplinary team.

### 6.3. Partial Breast Irradiation

Similar to the way in which a simultaneous or sequential boost requires adequate delineation of the lumpectomy site, so too does partial breast irradiation (PBI). PBI is an emerging option that can be used to treat patients with a conserved breast. The basic principles involve delivering radiation to the tumour cavity plus a wide margin using higher doses per fraction in a hypofractionated regime. Patients suitable for PBI include those with small, biologically favourable tumours and the benefit lies in minimizing dose exposure to surrounding normal tissue. In the literature, both the oncological and the cosmetic outcome results have been conflicting but PBI remains a suitable treatment option for some women [78,84,85,86].

PBI can be delivered via multiple methods including single-dose intra-operative radiotherapy, interstitial brachytherapy catheters or external beam radiotherapy, most of which is reported to be used in the non-oncoplastic surgical setting. There is a paucity of high-level evidence regarding the use of PBI in the oncoplastic setting.

Liu et al. recently published retrospective data assessing the efficacy and safety of oncoplastic surgery and intra-operative radiotherapy (IORT) in 32 patients. Patients were included if margins were negative on post-operative histopathological assessment and all patients had pT1-2N0-1 breast cancer. IORT was delivered with electron therapy to the gross tumour plus a 2 cm margin prior to excision. With a median follow up of 29 months, cosmesis was excellent and patient satisfaction was high using the SF-36 questionnaire. Oncological outcomes were in line with historical data. The authors concluded that IORT was safe in the setting of oncoplastic breast surgery and cosmetic outcome was improved [87]. It is important to note that the study included small number of patients and follow up is short. Additionally, IORT is not always practical, can be time consuming and as mentioned, has conflicting oncological and cosmetic outcomes in the literature.

Given the entire treatment volume is based around the tumour bed in PBI, accurate delineation in this situation is imperative to PBI to be safely implemented.

### 6.4. In the Post-Mastectomy Setting

In 1991, the National Institute of Health Consensus Conference endorsed breast-conservation therapy as the standard of care for early breast cancer [88]. This was based on evidence showing equivalence between mastectomy and breast conservation with adjuvant radiotherapy [19,20]. However mastectomy is often the best surgical approach for women with inflammatory breast cancer, skin involvement, large tumours in which conserving the breast would lead to poor cosmetic outcomes or recurrent positive margins. Additionally, there are many women who elect for mastectomy for various reasons including mutation positive patients (i.e., BRCA 1 or BRCA 2 mutations) or personal preference.

Reconstruction of the chest wall after mastectomy needs to be discussed with patients as the psychological distress of an unreconstructed breast can be significant for some patients [89]. There are reports that women having reconstruction after mastectomy report greater satisfaction with breasts, and psychosocial and sexual well-being compared to those with an unconstructed breast [90].

Reconstructive options need to be considered in the context of indications for adjuvant radiotherapy for reasons discussed below.

The indications for post-mastectomy radiotherapy (PMRT) continues to evolve but most consensus statements utilize well known literature data and a meta-analysis that confirmed a significant improvement in local control and overall survival in breast cancer patients with locally advanced disease or margin positivity. Whether patients with T1-T2 disease or 1-3 axillary nodes need PMRT remains an ongoing debate [60,91,92,93,94]. 

Despite the oncological advantages of PMRT, it has been reported to be associated with a moderate increase in complications like skin fibrosis, infection, distortion of the breast shape, volume loss, fat necrosis, implant malfunction which translates into poor cosmesis, and poor patient satisfaction [95,96,97,98].

Whether these outcomes are worse according to timing of reconstruction, be it immediate versus delayed procedure, has been controversial. Early series evaluating immediate autologous reconstruction described concerning outcomes with PMRT with increased incidence of late complications including fat necrosis, contracture, and volume loss compared with a delayed approach. Immediate autologous reconstruction is an extensive and delicate procedure and irradiating these tissues may result in higher complications with patients needing salvage procedures [99]. This led to many centers avoiding radiotherapy in a breast that has been immediately reconstructed with a preference to delay reconstruction until after PMRT. However, growing evidence suggests that autologous flap reconstruction may tolerate radiotherapy [100].

Outcomes also appear to vary according to the type of reconstruction performed. Patients with implant-based reconstruction and radiotherapy had lower rates of satisfaction at two years compared to those who had autologous reconstructions and radiotherapy. Jagsi et al. published a prospective cohort study that assessed the impact of radiotherapy on the reconstructed breast. Radiotherapy appears to increase breast complications and impair patient-reported satisfaction among patients receiving implant reconstruction when compared with autologous reconstruction. At two years, major breast complications occurred in 33.2% of irradiated patients receiving implant-based reconstruction, 17.6% of irradiated patients receiving autologous reconstruction, 15.6% of unirradiated patients receiving implant-based reconstruction, and 22.9% of unirradiated patients receiving autologous reconstruction [101]. Despite this, implant reconstruction and radiotherapy has been shown to be associated with good patient satisfaction and low decisional regret [102].

The landmark Mastectomy Reconstructive Outcomes Consortium (MROC) publication of complications in breast reconstruction did not see any significant effects on complications for the patient cohort receiving radiation before reconstruction. The group comments on the little consensus amongst previous studies on complication rates for breast reconstruction in irradiated patients but does mention that radiation increases the risk of complications in immediate/expander reconstructions which may be far less profound in autologous tissue procedures. After 2 years, among patients who had immediate reconstruction, there was at least 1 reported complication in 39% of patients who had radiotherapy compared to 22% of those who did not. Amongst patients who had autologous reconstruction, the rates of complications were similar regardless of radiation. On multivariate analysis, radiotherapy was associated with 2.6 higher odds of complications amongst those who had implant reconstruction (*p* < 0.001). Similarly, the rates of reconstruction failure at 2 years was higher in those who had implant reconstruction and radiotherapy while failures occurred in <5% of all other groups. Patient-reported outcomes with implant reconstruction were also reportedly worse in those who had implant reconstruction and radiotherapy [103].

The need for PMRT is not always established pre-operatively and may influence the ultimate surgical choice due to the reasons discussed above. As such, it is recommended that all women are discussed through the multidisciplinary team prior to undergoing mastectomy to determine the likelihood for PMRT. Cosmesis and patient reported outcomes are worse and does seem to vary according to the type of oncoplastic surgical procedure. Thus, it is imperative that patients understand the risks and benefits of adjuvant radiotherapy in the setting of oncoplastic surgery prior to decision making regarding the type of surgical procedure they will undergo [5].

### 6.5. Technical Considerations with Radiotherapy and the Reconstructed Breast

Early studies report that immediate breast reconstruction can result in compromised PMRT plans, particularly with left-sided treatment and coverage of the internal mammary chain [104].

This is largely due to the contour shape that can lead to an unacceptable increase in heart and lung doses which may impact on the delivery of PMRT [105,106].

Newer conformal techniques like Intensity-Modulated Radiotherapy (IMRT)/Volumetric Modulated Arc Therapy (VMAT) may be able to achieve better coverage of the internal mammary chain, improved dose homogeneity, and sparing of surrounding organs at risk while including the IMC in this setting [107,108].

Similarly, utilising immobilisation techniques like Deep-Inspiration Breast Hold (DIBH) may provide an additional benefit [109,110].

There is no convincing evidence that reconstructive surgery compromises oncological outcomes in relation to radiation efficacy, but the technical issues of radiation delivery, the timing and type of surgical procedure are critical considerations in obtaining the best management plan for patients which is best done through the multidisciplinary meeting [5].

### 6.6. Reverse-Sequencing Radiotherapy

Benefits of an immediate breast reconstruction include single operative procedure, shorter overall treatment time, and improved psychosocial well-being. As previously discussed, many specialists are reluctant to irradiate an immediately and permanently reconstructed breast.

Some groups have been exploring the option of pre-operative radiotherapy or commonly called “reverse-sequencing radiotherapy” where radiation is delivered prior to a reconstructive procedure [111]. This pathway is attractive as it means adjuvant therapies are not delayed and also avoids expander complications and the traditional two-stage reconstruction pathway. European groups are discussing clinical trials and a British group has started a prospective study of this “reverse-sequence” approach [112]. Promising single centre experiences have been published over the last decade, suggesting equivalence in oncological outcomes in chemotherapy responders and interest is growing [113,114,115,116]. The main barrier to this approach is in women with triple negative and Her-2 amplified cancers where locoregional histopathological response to neoadjuvant systemic therapy currently determines need for further adjuvant therapy and this is, potentially, lost after neoadjuvant radiotherapy [117,118].

### 6.7. Fractionation Schedule

Traditional dose and fractionation for breast radiotherapy has been 50Gy in 25 fractions however multiple trials over the last decade have shown equivalence for tumour control and survival with a moderately hypofractionated (40–42.56Gy in 15–16 fractions) approach. Additionally, acute and late effects have been shown to be similar, if not improved, with a hypofractionated schedule [119,120,121]. More recently, data has shown ultra hypofractionated whole breast radiotherapy (26Gy in 5 fractions) is non-inferior to hypofractionation in terms of local control and lower acute with similar late toxicities [122]. This is likely to be true for all patients, regardless of type of surgery as supported by the recent ESTRO guidelines of which 91.3% of the expert panel supported hypofractionated whole breast radiotherapy after oncoplastic breast-conserving surgery and 86.9% supported hypofractionation in a reconstructed chest wall of any type. Emerging data with long term follow-up is likely to continue to change practice.

## 7. The Role of Medical Oncology

### 7.1. Neoadjuvant Chemotherapy

Systemic chemotherapy is the standard of care for high-risk advanced and inoperable tumours with equivalent outcomes confirmed between neoadjuvant and adjuvant administration [123,124].

Systemic chemotherapy is increasingly prescribed for high-risk tumours based on prognostic and predictive gene assays that estimate the risk of recurrence and potential benefit of chemotherapy. [125]. Advantages of neoadjuvant therapy include early treatment of micrometastatic disease, downstaging tumours to minimise the extent of breast and nodal surgery and tumour histopathological response assessment to guide prognostication and further management [126,127].

The advantage of downstaging tumours may benefit patients by reducing requirement for mastectomy, oncoplastic breast surgery complexity, and volume of excision in patients suitable for breast conservation [16].

There appears to be no increased risk of surgical complications in those patients who have received neoadjuvant chemotherapy and proceed to surgery [128,129,130,131].

MROC data did not find any significant effects by chemotherapy on complications in breast reconstruction and the MD Anderson group has also published its results showing no increased risk of complications in oncoplastic breast surgery following neoadjuvant chemotherapy [103,132].

Despite the concern that delaying primary surgery with neoadjuvant chemotherapy may increase mortality [120] no difference in oncological outcomes has been observed when surgery is performed up to 8 weeks after neoadjuvant chemotherapy [128,133]. One major driving influence on increasing use of neoadjuvant chemotherapy is the demonstration of tumour sensitivity or resistance that tailors subsequent systemic therapy [117,118].

### 7.2. Adjuvant Chemotherapy

When imaging obtained after delivery of neoadjuvant chemotherapy suggests a residual cancer burden, adjuvant therapy may be indicated which may influence the type of surgery performed. Short-term complications are slightly higher with more complex oncoplastic surgery which may delay or hinder the ability to deliver adjuvant systemic therapy and impede long-term oncological outcomes [134]. Simpler surgery with a lower risk of infections or delayed wound healing may be preferable.

The addition of chemotherapy whether given in the neoadjuvant or adjuvant setting is an important part of the management paradigm for these patients. The optimal sequencing of surgery, radiotherapy, and delivery of chemotherapy may vary between patients and the involvement of all specialists at the multidisciplinary team is imperative for best outcomes.

## 8. Conclusions

Individualising the optimal sequencing and delivery of multidisciplinary care is complex for breast cancer patients and decisions should not be made in isolation. The introduction of oncoplastic surgery and the various scenarios in which neoadjuvant or adjuvant systemic therapy and/or radiation therapy adds to the complexity of the treatment paradigm. As always, excellent patient centred care should be the specialists’ main goal and it is clear that this is achieved through utilisation of a well-established multidisciplinary team. Robust high-level evidence for oncoplastic surgery is lacking but randomised data is unlikely to ever be produced for the majority of situations. However, it appears at least equivalent in oncological terms and superior in QOL outcomes. Whilst evidence is continuing to evolve, specialists should rigorously monitor and evaluate oncological, psychosocial, and aesthetic outcomes for the breast cancer patient.

## Figures and Tables

**Table 1 cancers-14-01685-t001:** Summary of oncoplastic surgery in breast-conservation setting.

**Level 1: Basic Techniques/Partial Breast Reconstruction**	
Description of Surgical Principles	Types of Commonly Performed Procedures
Used when up to 20% breast tissue removed Dual-plane undermining and re-approximation of tissues No skin excision or major nipple relocation needed	Can be used in any quadrant via: Skin incision Skin undermining Nipple-areolar complex (NAC) undermining Full thickness excision of tumour and margins Full thickness breast tissue reapproximation Minor NAC repositioning
**Level 2: Complex Techniques**	
Description of surgical principles	Types of commonly performed procedures
Derived from mastopexy procedures Can excise 20–50% breast tissue Often involves reshaping of breast tissue and reposition of NAC	Parellogram skin incision Lateral segmentectomy Batwing mastopexy Central breast resection Crescent mastopexy Reduction mammoplasty Donut mastopexy Vertical mammoplasty

**Table 2 cancers-14-01685-t002:** Summary of three most common autologous reconstructive options and their attributes.

Attributes	TRAM Flap	DIEP Perforator Flap	Latissimus Dorsi Flap
Summary	Transversely oriented ellipse of lower abdominal skin, fat and rectus abdominus muscle as donor site. Can be a rotation pedicle flap or free flap with vascular anastomosis to the breast site.	Transversely oriented ellipse of lower abdominal skin and fat as donor site (spares rectus abdmoninis muscle). Most common free flap procedure used for breast reconstruction. Dependent on number, calibre and location of perforating vessels	Pedicled muscular/aponeurotic flap. Flap is tunnelled from Latissimus Dorsi through axilla Often utilised in combination with implant for added breast volume. Robust blood supply More likely to require revision surgery long term compared with abdominal free flaps.
Size constraints	Average weight Adequate abdominal soft tissue (to match volume loss of breast). Large volume reconstructions can be achieved.	Adequate abdominal soft tissue (to match volume loss of breast). Achieves less volume than TRAM reconstruction.	Non-weight dependent. Limited breast volume when LD flap used alone. Muscle can atrophy with time.
Healing	Patient selection critical to avoid flap loss + donor site morbidity. Non-smoking. Non-diabetic or well-controlled diabetes. Longer operating times. Longer hospital stay. Mesh used to repair abdominal muscle defect. Can cause chronic pain.	Patient selection critical to avoid flap loss + donor site morbidity. Non-smoking. Non-diabetic or well-controlled diabetes. Lower incidence abdominal wall laxity and weakness. Acceptable recovery time.	Low incidence of flap loss due to good blood supply to muscle and no microvascular reconstruction needed. Donor site scarring can be an issue. Latissimus dorsi muscle can atrophy over time, causing contour irregularities.
Muscle-function loss	Possible deficits in abdominal flexion and extension.	Improved abdominal muscle function with total muscle preservation.	Can impact shoulder function.

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
