# Peer review of "Breast Radiotherapy after Oncoplastic Surgery—A Multidisciplinary Approach"

_cancers, 2022, doi:10.3390/cancers14071685_

Round 1

Reviewer 1 Report

Overall well written and comprehensive.  Controversies are addressed.

Minor comments:

  1. p3 line 98: "prior" should read "prior or planned".
  2. The section on 3.2.2.2 could add a table of attributes for the various flaps; to include size constraints, shape, technical difficulties, healing differences, and loss of muscle functions for the patient.
  3. The section of 3.2.2.3 does not discuss the frequent and rather distressing consequence of fat necrosis on the texture and contours when using fat grafting.  Personally, I have seen quite a few poor patient satisfaction outcomes with fat grafting.
  4. page 12 line 544 (and others): please define acronyms when first used (here is MROC).  DIBH and others.
  5. Not mentioned is the additional OR time needed for the oncoplastic procedure, particularly with the frequent follow-up procedures to adjust cosmesis.
  6. Is there any literature to deal with patients who develop keloidal scarring?

Reviewer 2 Report

The title of the manuscript refers to radiotherapy after oncoplastic surgery and the role of a multidisciplinary approach. Oncoplastic surgery is well defined and this part of the manuscript is clear. However, the part describing the role of the multidisciplinary team is not specific and concrete enough. The section describing radiotherapy is also insufficiently specific and concrete: it does not include specific, scientifically based suggestions and dilemmas for tumour bed delineation (e.g. the specific role of preoperative imaging, the possible role of preoperative chemotherapy, etc.), which is a very important open question after conservative oncoplastic surgery.  I miss the pro et contra approach that should be an integral part of every review article.
-Chapter 6.2., title: Boost: I suggest replacing with "dose addition to the tumour bed".

Round 2

Reviewer 2 Report

The manuscript has been completed, but the added text is mostly of no significant scientific value. The additional text on the multidisciplinary team (lines 352-367) is without one reference cited.